# Implications of Physical Exercise on Episodic Memory and Anxiety: The Role of the Serotonergic System

**DOI:** 10.3390/ijms241411372

**Published:** 2023-07-12

**Authors:** Ricardo Illesca-Matus, Nicolás M. Ardiles, Felipe Munoz, Pablo R. Moya

**Affiliations:** 1Laboratorio de Neurodinámica Básica y Aplicada, Escuela de Psicología, Pontificia Universidad Católica de Chile, Santiago 8320000, Chile; 2Centro de Investigación Avanzada en Educación (CIAE), Universidad de Chile, Santiago 8320000, Chile; 3Centro Interdisciplinario de Neurociencia de Valparaíso, Facultad de Ciencias, Universidad de Valparaíso, Valparaíso 2340000, Chile; nicolas.ardiles@postgrado.uv.cl; 4Programa de Doctorado en Ciencias e Ingeniería para la Salud, Universidad de Valparaíso, Valparaíso 2340000, Chile; hernan.alvarez@postgrado.uv.cl; 5Instituto de Ciencias de la Salud, Universidad de O’Higgins, Rancagua 2820000, Chile; 6Instituto de Fisiología, Facultad de Ciencias, Universidad de Valparaíso, Valparaíso 2340000, Chile

**Keywords:** serotonin, physical exercise, anxiety, episodic memory

## Abstract

There is a growing interest in investigating the effects of physical exercise on cognitive performance, particularly episodic memory. Similarly, an increasing number of studies in recent decades have studied the effects of physical activity on mood and anxiety disorders. Moreover, the COVID-19 pandemic has raised awareness of the importance of regular physical activity for both mental and physical health. Nevertheless, the exact mechanisms underlying these effects are not fully understood. Interestingly, recent findings suggest that the serotonergic system may play a key role in mediating the effects of physical exercise on episodic memory and anxiety. In this review, we discuss the impact of physical exercise on both episodic memory and anxiety in human and animal models. In addition, we explore the accumulating evidence that supports a role for the serotonergic system in the effects of physical exercise on episodic memory and anxiety.

## 1. Introduction

Mounting evidence substantiates the notion that physical exercise promotes and protects both mental and physical health [1,2,3]. Regular physical activity constitutes a pivotal protective factor for the prevention and management of noncommunicable diseases (NCDs) [4]. Consequently, physical inactivity and sedentary lifestyles have emerged as major risk factors for the onset of NCDs. The global economic burden resulting from physical inactivity is striking: an estimated nearly 500 million new cases of preventable NCDs are projected to arise between 2020 and 2030, incurring annual treatment costs of approximately USD 27 billion if the prevailing prevalence of physical inactivity remains unchanged [4,5]. Physical exercise has a highly influential role in the prevention of type 2 diabetes [6] and reducing the risk of cardiovascular disease [7,8]. Moreover, regular exercise enhances cognitive functions and memory in middle-aged and elderly individuals [9,10], as well as in children and adolescents [11,12]. Numerous studies have demonstrated that regular exercise not only improves memory and learning, but also provides protection against future memory problems in the elderly [13] and those resulting from neurodegenerative disorders [14,15,16]. Furthermore, physical activity has been observed to mitigate the disabling symptoms of neuropsychiatric disorders. Both patients with post-traumatic stress disorder and bipolar disorder exhibited decreased depressive symptoms and improved sleep quality after physical exercise [17,18]. Additionally, voluntary chronic exercise improved psychiatric symptoms and the quality of life in patients with schizophrenia [19,20].

Nevertheless, the effects of exercise on the interplay between physical, mental, and cognitive conditions have received relatively less exploration. Investigating these interactions is pertinent not only to unveil potential correlations, but also to ascertain causal mechanisms that hold relevance for the realms of health and education. In this regard, although the individual effects of exercise on anxiety and episodic memory are well documented, the underlying mechanisms and their potential interplay remain far from being fully understood.

Several studies have demonstrated impairments in memory performance associated with mood and anxiety disorders [21,22,23,24]. Recent findings indicate that physical exercise not only alleviates specific symptomatology of these disorders, but also improves memory disturbances linked to them [25,26]. Consequently, the manifold benefits of physical exercise on cognitive performance and symptomatic relief in mood and anxiety disorders may rely on common cellular and molecular mechanisms. One extensively studied exercise-induced aspect is brain-derived neurotrophic factor (BDNF), which exhibits elevated levels following long-term physical exercise in both animal models and humans [27,28]. Additionally, Pietrelli and colleagues reported that aerobic exercise promotes increased levels of the neurotransmitter serotonin (5-HT) in the raphe nuclei and a greater number of mature neurons in the hippocampus of rats, changes that were associated with enhancements in spatial and non-spatial memory [29]. Considering the predominantly distinct evidence on the role of 5-HT in emotional regulation as well as in memory processes, this review delves into the role of the serotonergic system in modulating the effects of physical exercise on both episodic memory and anxiety. With this aim in mind, we start by elucidating the concept of episodic memory.

## 2. Episodic Memory

Memory plays a crucial role in our daily lives by allowing us to store and retrieve information at different moments, and it can be primarily divided into two types: declarative memory, which involves conscious access to information, and non-declarative memory, which is expressed through behavioral changes without the need for awareness. Within declarative memory, we find semantic memory, related to general knowledge, and episodic memory, which is defined as “the memory of memories about events in a specific context and time” [30]. Episodic memory also involves the ability to imagine or pre-experience events that may potentially occur in the future [31], known as prospective memory, and both are closely interconnected even sharing the involvement of critical structures for their functioning, such as the medial prefrontal cortex (mPFC), posterior medial parietal cortex, and the medial temporal lobe (MTL) [32]. This means that episodic memory not only allows us to learn, store, and retrieve information about our unique personal experiences, which is fundamental for our understanding of the past, but also to anticipate future events and plan actions in imagined scenarios that may happen in the future.

Importantly, the discovery of the brain networks underlying episodic memory largely emerged from studies of human cases in neurology, neuropsychiatry, and neuropsychology, in which the investigations of famous patients by Brenda Milner and others since the 1950s were particularly significant. Thanks to the accumulated evidence from these studies to date, we know that episodic memory is supported by a large network of brain areas, including neocortical association areas and components of the medial temporal lobe (MTL), including the parahippocampal cortical areas and the hippocampus [33]. The anatomical organization of the key pathways of interaction between the neocortex and medial temporal areas, as well as the organization of the medial temporal areas themselves, are largely conserved across mammalian species, from rodents to primates [34].

Crucially, disorders affecting episodic memory can lead to amnesic syndromes in humans, which is particularly evident in neurodegenerative disorders such as Alzheimer’s disease [35] and conditions such as hippocampal sclerosis [36]; in some neuropsychiatric disorders, such as depression [37] and post-traumatic stress disorder [38], memory systems, including episodic memory, also seem to be affected. Interestingly, despite being strongly related to these neuropsychiatric disorders, the role of anxiety in the decline in episodic memory has only recently been investigated [39].

## 3. Anxiety

Anxiety emerged as a complex emotional state characterized by the apprehension of future threats [40]. With evolutionary roots, anxiety can trigger adaptive reactions in specific contexts, serving to enhance the survival of the organism by facilitating the avoidance of potentially hazardous situations [41]. Nevertheless, when these responses become excessively intense and/or persist over time and significantly impair an individual’s overall well-being, they can give rise to mental health disorders [40]. Crucially, anxiety stands as the most prevalent psychiatric disorder worldwide [42], impacting an estimated 3.6% of the global population [43]. Consequently, anxiety disorders incur substantial costs for both individuals and society. In the case of adults, selective serotonin reuptake inhibitors (SSRIs) are frequently recommended as the initial line of treatment for anxiety [44,45]. Nevertheless, existing pharmacological approaches for anxiety exhibit certain limitations, such as a high financial burden, delayed onset of therapeutic effects, limited clinical efficacy, unwanted side effects, and the associated stigma surrounding drug usage and dependency [46,47].

We know that anxious individuals have negative beliefs and cognitive avoidance behavior because of their interpretation of internal and external stimuli as highly dangerous, due to the bias of selectively retrieving personal relevant information from the past that would confirm their negative interpretation of current or anticipated situations [48,49,50]. This negativity bias is accompanied by an intolerance of uncertainty about the future [51] and, crucially, by impairments in episodic memory for neutral and emotionally irrelevant information [21].

Interestingly, recent research supports the hypothesis that the presence or severity of anxiety is associated with a lower cognitive performance, particularly in older adults. These effects would be particularly pronounced in episodic memory compared with other cognitive domains [52]. Therefore, we will continue detailing the research that reports the detrimental effects of anxiety on episodic memory.

## 4. Episodic Memory and Anxiety

A growing body of evidence has reported the effects of anxiety on episodic memory. For example, Payne et al. [53] induced psychosocial stress prior to the encoding of neutral or emotional images with narration. Stress manipulation also increased indicators such as salivary cortisol and subjectively reported stress, which were further related to anxiety levels. Interestingly, stressed subjects reported more false memories than non-stressed control subjects, and these false memories were positively correlated with cortisol levels, providing evidence of a relationship between stress, associated anxiety, and false memory formation. Specifically, Noël et al. [54] observed that individuals with transient global amnesia exhibited higher levels of anxiety and more depressed moods than controls; this alteration in emotional state correlated with deficits in episodic memory. Along the same lines, Lachman and Agrigoroaei [55] found that lower levels of control beliefs were associated with higher anxiety levels, which, in turn, affected episodic memory performance by increasing the likelihood of interference in a task, in which age, gender, and verbal abilities were covariates. Furthermore, Moscovitch et al. [56] observed that while individuals with low social anxiety were able to access a relatively balanced variety of positive and negative self-representations rich in episodic details, individuals with high social anxiety retrieved a higher and imbalanced proportion of negative images and memories, as well as poor positive images that had significantly degraded in episodic detail. Finally, negative images influenced the two groups differently, and individuals with high social anxiety experienced more negative emotional and cognitive consequences associated with bringing such images to mind.

In a 3-year follow-up study of older adults, Fung et al. [39] explored how anxiety symptoms were associated with memory decline in cognitively unimpaired older adults. Anxious healthy older adults showed a specific decline in episodic memory over a 3-year interval. On the other hand, Cansino et al. [57] identified factors that act together as mediators of episodic memory decline throughout adulthood. One of the mediators found was anxiety, which has negative effects on memory accuracy and influences memory performance because other processes interfere by requiring attention or resources intended for memory representations. These results suggest that anxiety may mediate or predict episodic memory decline, and these effects are most evident in older adults, even when they are cognitively healthy.

Interestingly, the impact of anxiety and worry on episodic memory, two previously unstudied states that are strongly linked, has also been explored. In a series of studies, Pajkossy et al. [58,59] found that trait anxiety and trait worry exert a partially opposing effect on free recall performance. This means that higher levels of worry are associated with better performance in episodic memory tasks that require strategic and effortful retrieval, while higher levels of anxiety are associated with poorer performance in the episodic memory task.

Procedures to improve performance on episodic memory tasks have been investigated, which may also influence anxiety, such as Autobiographical Episodic Memory-based training and the induction of episodic specificity (for more details, see [60,61]); however, other types of interventions, such as exercise, have been reported to improve cognitive performance in general, including episodic memory, as well as having anxiolytic effects. Advancing the understanding of how exercise achieves these effects is a promising field of research. Next, we will present the effects of exercise on episodic memory.

## 5. Exercise and Episodic Memory

The influence of exercise on memory has predominantly been examined in the spatial domain [62], which falls outside the scope of this review (for interested readers, see [63]). Conversely, the precise effects of exercise on episodic memory have only recently started to be investigated. However, the integration of these three components in episodic memory tasks (event, time, and space) has posed challenges impeding the study of their relationship with exercise. Moreover, the intensity and mode of exercise, along with the physical condition of the participants, have also presented challenges for researchers. Next, we will review studies of the effects of exercise on episodic memory (Table 1).

### 5.1. Clinical Studies about Exercise and Episodic Memory

Early studies in this field primarily focused on the “event” aspect of episodic memory and examined the short- and long-term recall of words using a word recall paradigm [64,65,66,67]. However, some studies did not find any significant effects. For instance, when exercise sessions were set at a moderate intensity of 70% of the participant’s heart rate, no impact of exercise on short- and long-term word recall in episodic memory was observed [68]. Similarly, in a study evaluating the combined effects of acute exercise and the Episodic Specific Induction (ESI) paradigm on episodic memory, assessed through an autobiographical memory task and the Treasure Hunt Task (THT) (which assesses the three components of episodic memory: “What”, “Where”, and “When”), no exercise-related effects on episodic memory were found [69]. A meta-analysis [70] reported that mixed interventions involving both aerobic and resistance training tend to produce the most pronounced effects on episodic memory. However, many of these investigations did not consider the impact of vigorous intensity and rest period between exercise and the memory task, which have been shown to modulate the effect of acute aerobic exercise on episodic memory [71].

Several investigations have used acute exercise on a treadmill and stationary bicycle, typically lasting for 30 min at a moderate intensity. This level of physical activity has been shown to improve short- and long-term word recall [72,73,74,75]. Furthermore, an increase in recall has been observed by presenting lists multiple times [76], while episodic false recall in short- and long-term memory has decreased [77,78]. Additionally, both moderate- and vigorous-intensity interventions have demonstrated positive effects on episodic memory when a longer time interval is present between exercise and the immediate task (15 min of recovery) as well as the long-term episodic memory task (24 h post exercise) [79].

Another study by Suwabe and colleagues investigated whether ten minutes of moderate-intensity acute exercise (50% of VO_2_ peak) increased hippocampal activity, particularly in the dentate gyrus (DG), leading to enhanced pattern separation, a critical component of episodic memory [80]. Interestingly, moderate-intensity acute exercise in humans was found to improve mnemonic discrimination, suggesting that it may enhance DG-mediated pattern matching. Furthermore, in a study involving ten minutes of acute exercise at a light intensity, combined with high-resolution functional magnetic resonance imaging techniques, it was demonstrated that exercise improved pattern separation and increased functional connectivity between the DG/CA3 hippocampus and cortical regions (parahippocampal region, angular region, and fusiform gyri) [81]. Importantly, the magnitude of improved functional connectivity predicted the degree of memory enhancement at the individual level, suggesting that acute light-intensity exercise can rapidly enhance hippocampal memory function. This improvement may be attributed to increased functional DG/CA3-neocortical connectivity, which could also underlie the effects of exercise on episodic memory.

Acute resistance exercise performed before encoding has been observed to enhance the “where” component of episodic memory tasks in both group settings [82] and within-subject experimental design, with moderate demand levels preceding the Rey Auditory Verbal Learning Test (RAVLT) [83]. Likewise, acute resistance exercise involving knee extension/flexion during the consolidation period of an episodic memory task has demonstrated positive effects, particularly when the memory task has affective valence [84]. However, these effects were not observed when exercise was conducted during the consolidation period [85] or when the exercise intensity was vigorous [86]. Moreover, studies utilizing a research design with four experimental groups have not shown exercise before encoding to be significantly superior to the control group [87]. Additionally, the impact of exercise has not been observed in similar experimental designs involving treadmill exercise sessions with progressively increasing intensity until reaching the maximum intensity for the participants [88].

The effects of chronic aerobic and resistance training on components of episodic memory, particularly using the RAVLT, have also been investigated. The most substantial effects have been associated with aerobic training [89], primarily observed in interventions lasting longer than 24 weeks [90,91,92]. Furthermore, a correlation has been observed between VO_2_ max and improved episodic memory performance [93], as well as between accelerometer-measured physical activity and episodic memory in healthy older adults [94], but not in older adults at risk for Alzheimer’s disease (AD) [95]. The combination of chronic exercise (aerobic, resistance, and sometimes stretching) has shown promising results. In older adults, a combined exercise strategy was implemented three times a week for four weeks, with a moderate to vigorous intensity [96]. The results of this protocol demonstrated a positive influence of exercise on a wide range of tests assessing episodic memory. Similar findings have been observed in the long-term episodic memory components when using combined exercise compared with exclusive aerobic exercise. Comparable results have been found in older adults with dementia [97] and adults with mild cognitive impairment [98].

### 5.2. Pre-Clinical Studies about Exercise and Episodic Memory

In research utilizing animal models, episodic memory is commonly evaluated through tasks involving pattern matching, as well as recognition and memory tasks involving novel places and objects [99]. Additionally, aerobic exercise in rodents has been demonstrated to be a potent stimulator of cell proliferation in the dentate gyrus, which may play a role in the proper functioning of memory. For instance, rodents that engage in running wheel activity have shown improved memory in recognizing objects and places. One of the mediators of this effect is bone morphogenetic protein (BMP) signaling, a protein that facilitates the impact of exercise on neurogenesis in the adult hippocampus. This signaling mechanism has been proposed as a shared mechanism underlying the effects of physical exercise on recognition, spatial, and associative memory [100]. Accumulating evidence supports the protective effects of exercise in affective and cognitive domains. A study by Schoenfeld and colleagues [101], using a 6-week exercise protocol, revealed that both mild and intense exercise effectively reversed the decrease in the survival of newborn cells in the hippocampus induced by chronic stress and prevented cognitive decline.

Furthermore, in mice genetically modified to down-regulate adult hippocampal cell proliferation, exercise rescued the generation of hippocampal cells, enhanced dendritic arborization, and compensated for recognition memory deficits [102]. Similarly, swimming exercise has been shown to prevent long-term memory impairment in object recognition caused by arsenic exposure, and this protective effect of physical exercise may be mediated by BDNF and cAMP response element-binding protein (CREB) in the dorsal hippocampus [103]. In a mouse model of chronic inflammatory pain, exercise reversed the downregulated of BDNF levels in the dentate gyrus and had performance-enhancing effects on recognition memory [104]. Martínez-Drudis and colleagues found that treadmill exercise can reverse impaired performance in recognition memory in rats with traumatic brain injury, particularly in the aspects related to ‘when’ and ‘where’ components [105]. Additionally, hippocampal BDNF levels were positively correlated with exercise and memory of ‘when’ (but not ‘where’). Taken together, these results suggest that modulators of exercise effects on episodic memory, such as hippocampal BDNF, may be more closely associated with certain memory components than others [105]. Lastly, it has been observed that in rodents with closed head injury, early moderate-intensity exercise effectively restores object recognition memory and prevents progressive neuronal loss. It also induces microglia activation through the restoration of BDNF expression and mitogen-activated protein kinase phosphatase-1 (MKP-1) in the hippocampus [106].

Specific types of exercise have been reported to have different effects on brain activity and cognition. For instance, differences in hippocampal volume have been observed between aerobic exercise and resistance exercise [107]. Unlike aerobic exercise, resistance exercise does not increase resting serum BDNF levels [108]. In rodents, both aerobic and resistance training upregulate protein levels associated with synaptic plasticity, such as synapsin 1 and synaptophysin. Despite the different signaling pathways activated depending on the type of exercise, both aerobic and resistance training improve spatial learning and memory [109]. Furthermore, treadmill exercise has been shown to increase the expression of proteins such as IGF-1, BDNF, TrkB, and calcium/calmodulin-dependent kinase II (b-CaMKII) in the hippocampus. Additionally, resistance training increases the levels of IGF-1 in the periphery and the hippocampus, as well as activates its receptor signaling pathway (Akt protein) in the hippocampus [62]. This evidence suggests that the activation of different signaling pathways by aerobic and resistance exercise could underlie the distinct effects on episodic memory. Similar physiological effects of both types of exercise may be involved in the effects of combined exercise on episodic memory. Despite studies suggesting that resistance exercise might enhance, but not replace, the effects of aerobic exercise on memory, future experiments will be necessary to clarify the mechanisms underlying this phenomenon.

## 6. Exercise and Anxiety

Human studies have demonstrated that physical activity has beneficial impacts on mood and anxiety across different age groups and genders [110,111]. Furthermore, research conducted on rodent models has demonstrated the anxiolytic and antidepressant effects of exercise [112,113]. Regular physical activity possesses both preventive and therapeutic properties for mood disorders [114,115]. This finding has captured the interest of neuroscientists, as physical exercise could serve as a valuable model to explore the underlying biological mechanisms of mental health disorders.

Mounting evidence suggests that incorporating exercise during adolescence may have long-term benefits in reducing anxiety and depression in adulthood [116,117,118]. From an evolutionary perspective, the transition from adolescence to adulthood represents a critical period of brain development and behavioral changes in both humans and rodents [119,120]. Regular exercise has also demonstrated the ability to restore normal functioning of the hypothalamic–pituitary–adrenal (HPA) axis [121]. Next, we will review studies evaluating the anxiolytic effects of exercise (Table 1).

### 6.1. Pre-Clinical Studies on Exercise and Anxiety

Numerous studies have consistently demonstrated that rodents engaging in voluntary and sustained running display notable improvements in behaviors associated with depression and anxiety [112,122,123]. For example, adult mice provided unrestricted access to running wheels for a period of three to four weeks demonstrated a decreased immobility time in the forced swim test (FST) [112,122,123]. Similarly, investigations focusing on 2 to 4 weeks of running in adult male C57BL/6 wild-type mice revealed that exercise increased center crossings and the time spent in the center of the open field test [112,113,122,123]. These findings suggest that chronic running effectively reduces anxiety-like behaviors in adult male mice. Importantly, the positive effects of exercise on anxiety-like behaviors can also extend to aging animals, as reported by Pietrelli and colleagues in a study involving 18-month-old rats that displayed a reduced anxiety-like phenotype in the elevated plus maze (EPM) [124]. Furthermore, the authors found that chronic running improves cognitive performance in older animals, as evidenced by a higher number of correct entries in the radial maze test [124]. Taken together, these findings highlight the therapeutic potential of chronic wheel running in alleviating anxiety- and depression-like behaviors in adult mice and rats, with these effects persisting in older animals.

### 6.2. Clinical Studies about Exercise and Anxiety

Extensive research conducted on humans, encompassing both healthy individuals and those with specific mental disorders, has revealed significant behavioral and neuropsychological changes associated with regular physical activity. In a diverse patient sample encompassing panic disorder, generalized anxiety disorder, and social phobia, a home-based walking program was found to enhance the clinical effectiveness of group cognitive behavioral therapy when compared with educational sessions focused on healthy eating [125]. The study by Merom and colleagues emphasizes the potential benefits of combining cognitive behavioral therapy with a physical activity regimen to improve clinical outcomes.

Furthermore, age-specific investigations have also revealed associations between anxiety and exercise. Among individuals aged 65 and older, it has been demonstrated that engaging in sports significantly reduces anxiety levels in the elderly population. Consequently, integrating a regular exercise plan into elderly care programs may be considered beneficial [126]. Similarly, in young adults aged 18 to 40, a randomized controlled trial indicated that a resistance exercise training program designed in accordance with WHO and ACSM guidelines reduced anxiety symptoms [127].

Considering the population from a broader perspective, including age and ethnicity, a meta-analysis comprising 14 cohorts from 13 unique prospective studies revealed that individuals reporting higher levels of physical activity had reduced odds of developing anxiety compared with those with lower physical activity levels. High self-reported physical activity was found to protect against the onset of agoraphobia and post-traumatic stress disorder, independent of ethnic background. Protective effects against anxiety were evident across Asia and Europe for children, adolescents, and adults [128].

Collectively, these studies support the notion that exercise may provide a protective effect against the onset of anxiety, regardless of demographic factors. Particularly, higher levels of physical activity have been shown to guard against agoraphobia, post-traumatic disorder, and various anxiety disorders. This highlights exercise as a promising, cost-effective, and easily accessible treatment option for individuals affected by these conditions.

## 7. The Role of the Serotoninergic System on the Effects of Exercise on Episodic Memory and Anxiety

Several brain pathways and systems involved in the impact of exercise on episodic memory have been proposed [82,129,130,131,132]; however, the mechanisms that can simultaneously influence episodic memory and anxiety in response to exercise have been less explored (Table 2). In this regard, there is evidence of concurrent effects of exercise on both episodic memory and anxiety. For example, a study reported that 4 weeks of voluntary exercise every other day improved object recognition memory and decreased anxiety-like behavior in Long Evans rats [133]. BDNF expression in the perirhinal cortex of exercising rats was strongly correlated with object recognition memory. Additionally, moderate-intensity exercise during the neonatal period in Wistar rats affected episodic memory and anxiety-like behavior, reducing anxiety, and improving episodic memory in adulthood [134]. However, these effects were not observed in later life. In a streptozotocin (STZ)-treated AD mouse model, swimming exercise led to a decrease in anxiety-like behavior and an improvement in recognition memory, accompanied by increased BDNF levels and decreased glutamate and TNF-α levels in the hippocampus [117].

**Table 1 ijms-24-11372-t001:** Preclinical and clinical studies that have investigated the effects of exercise type on episodic memory and anxiety. N.D., not determined.

Type of Study	Type of Exercise	Effect on Episodic Memory	References	Effect on Anxiety	References
Preclinical	Aerobic	Improvement	[100,102,103,105,106,107,117,122,133,134]	Anxiolytic	[101,104,106,112,113,117,121,123,124,133,134]
	Resistance	Improvement	[64]	N.D.	
Clinical	Aerobic	Improvement	[67,68,69,72,73,74,75,76,77,78,79,80,81,83,87,88,89,92,93,94,96,97,98]	Anxiolytic	[125]
	Resistance	Improvement	[71,84,91,93,96,97,98]	Anxiolytic	[127]

The neurotransmitter serotonin (5-HT) is an interesting candidate for mediating the simultaneous effects of exercise on anxiety and episodic memory [135,136]. Although the role of the dorsal raphe nucleus (DRN) (which contains a large proportion of 5-HT neurons) in the anxiolytic effects of exercise are increasingly studied, current evidence on its role in the effects of exercise on episodic memory is speculative. For instance, acute low-speed treadmill exercise in male Wistar rats was found to decrease anxious behavior and increase *c-fos* expression in 5-HT neurons in the DRN [137]. This suggests that this type of exercise can efficiently induce neuronal activation with anxiolytic effects.

Furthermore, studies have investigated the effects of treadmill exercise on anxiety and 5-HT expression in the DRN. Wang and colleagues [138] studied the effects of treadmill exercise (30 min, once daily for 10 days, starting on postnatal day 21 in a rat model of anxiety induced by maternal separation: physical activity was reported to reduce anxiety levels as well as to increase the expression of 5-HT and tryptophan hydroxylase (TPH) in the DRN. Furthermore, it enhanced GSK3β phosphorylation and suppressed β-catenin phosphorylation in the hippocampus [138]. Correspondingly, deregulated GSK3β activity has been reported along with altered serotonergic activity in mood disorders [139,140]; it has also been associated with anxiety and serotonin-sensitive social behavior [141]. On the other hand, the transcription factor β-catenin is considered a marker of GSK3β inactivation, since β-catenin levels in the cytoplasm are increased when GSK3β is inhibited [142]. Similar results were found in a swimming exercise protocol of 30 min per day for 4 weeks. This protocol was applied in rats to which anxiety was induced during old age through social isolation. Compared with the control group, exercise showed protective effects against anxiety in the elevated plus-maze test. In addition, the expression of TPH and 5-HT in the DRN was increased [143].

Exercise also modulates 5-HT levels and reduces anxiety levels at physiologically particular stages, such as pregnancy in rats. A study applied a protocol of exercise on a treadmill to pregnant rats and found that it had a protective effect against anxiety induced by a predator exposure protocol. Stressed pregnant rats showed increased 5-HT and TPH expression in the DRN, which was reduced by exercise during pregnancy [144]. This protocol had a protective effect against anxiety induced by a predator exposure protocol. This protocol was applied in a closed room during three daily 10-min sessions separated by 1 h. Stressed pregnant rats showed increased 5-HT and TPH expression in the DRN compared with non-stressed rats. Importantly, exercise during pregnancy reduced the expression of 5-HT and TPH.

The role of the DRN and its 5-HT neurons in the effects of exercise on episodic memory is not yet well-established. However, a study investigating the effects of 3-week treadmill exercise on the DRN found that exercise induced synaptic adaptations and enhanced persistent inward currents (PICs) in DRN 5-HT neurons, suggesting increased excitability and a potential mechanism underlying cognitive improvements such as episodic memory [145]. Nonetheless, more studies are needed to establish conclusive relationships between DRN activity and episodic memory, considering the electrophysiological, morphological, and neurochemical differences of the neurons and subfields within the DRN [146].

In contrast, it has been demonstrated that the expression of 5-HT receptors in key brain areas also plays a role in modulating the anxiolytic effects of exercise. For instance, a study using protocol involving voluntary exercise wheels for 6 weeks, which reduced anxiety-like behavior in rats, examined the role of 5-HT_2C_R in the basolateral amygdala (BLA) and dorsal striatum (DS). These two brain regions have been reported to be crucial in this type of behavior [147]. Additionally, the authors observed that the exercised rats showed improved fear response and learning abilities. Furthermore, the exercised rats exhibited lower levels of 5-HT_2C_R mRNA in both the BLA and DS. This indicates that the expression of 5-HT_2C_R mRNA in specific brain areas is influenced by the physical activity level of the organism. This suggests that 5-HT_2C_R could be a target for the beneficial effects of physical activity on anxiety. Interestingly, the stimulation of 5-HT_2C_R in the BLA has been shown to promote the induction of long-term potentiation in the dentate gyrus, which is a brain region involved in learning and memory processes [148].

Although no specific research has been conducted to date, some evidence suggests that 5-HT_2C_R may also play a role in the effects of exercise on episodic memory. One study investigated the effects of the 5-HT_2C_R antagonist RO 60-0491 on recognition memory in rats using the object recognition task [149]. The administration of RO 60-0491 after training, at a dose of 3 mg/kg, counteracted the rats’ performance deficits on the object recognition task, indicating that 5-HT_2C_R modulates information storage and/or retrieval. Furthermore, the effects of the nonselective 5-HT_2C_R antagonist mesulergin and the 5-HT_2C_R agonist mCPP (metachlorophenylpiperazine) on learning acquisition, short-term memory, and long-term memory were investigated using the Morris Water Maze Test [150]. The administration of mCPP altered all three types of memory evaluated, while rats injected with mesulergin did not show impaired memory functions. Additionally, when mCPP was injected, it did not induce memory deficits. These findings suggest that the inhibitory activity at 5-HT_2C_R could be a common mechanism underlying the effects of exercise on memory and anxiety. However, further investigations with specific tests to assess episodic memory and the identification of the specific brain areas involved in this modulation are required.

The modulatory role of 5-HT_2A_R has also been investigated. In a mouse model of anxiety induced by chronic stress, a treadmill exercise regimen consisting of 60 min per day, 6 days a week, for 21 days resulted in an anxiolytic effect, as measured by the open field test and elevated plus maze [151]. Interestingly, exercise improved 5-HT_2A_R synaptic density in BLA. The investigators further observed that the administration of the selective 5-HT_2A_R antagonist MDL11930 generated an anxiety phenotype. It is possible, then, that the synaptic recruitment of 5-HT_2A_R in BLA neurons is a mediator of the anxiolytic effects of exercise. In addition, exercise reduced adenosine A_2A_ receptor (A_2A_R)-mediated protein kinase A (PKA) activation. The anxiolytic effect of exercise was mitigated by local activation of A_2A_R within the BLA using CGS21680, a selective A_2A_R agonist. Thus, A_2A_R-mediated PKA activity was shown to be dependent on 5-HT_2A_R signaling in the BLA. These results imply that repeated stress increases A_2A_R-mediated adenosine signaling to facilitate PKA activation, while regular exercise inhibits A_2A_R function by increasing 5-HT_2A_R in the BLA. Consequently, this integrated modulation of 5-HT and adenosine signaling, via 5-HT_2A_R and A_2A_R, respectively, may be a mechanism underlying the anxiolytic effect of regular exercise [147]. Interestingly, in mice, (4-bromo-3,6-dimethoxybenzocyclobuten-1-yl) methylamine hydrobromide (TCB-2), a selective 5-HT_2A_R agonist, enhances object memory consolidation when administered systemically during a behavioral test for working memory [152]. However, studies specifically evaluating the role of 5-HT_2A_R in episodic memory are needed to establish conclusive relationships.

Interestingly, anxiolytic effects have also been observed in rats whose mothers received an exercise protocol. Specifically, maternal exercise caused a decrease in anxiety-like behavior in offspring. Crucially, the administration of 5-HT_2_ and D2 antagonists inhibited the protective effects of exercise and increased anxiety in offspring. This suggests that maternal exercise has a protective effect against anxiety in children. This effect could probably be mediated by the 5-HT_2_ and D2 receptors [153]. In addition, it is able to selectively prevent MDMA-induced verbal memory impairment [154]. This evidence on the possible role of 5-HT_2_R should be investigated to determine whether 5-HT_2_R is a candidate in modulating the effects of exercise on episodic memory, as has been shown in anxiety.

The role of 5HT_1A_ in the anxiolytic effects of exercise has also been investigated. A treadmill exercise protocol consisting of 19 m/min for 60 min/day, 5 days/week, from 0 to 8 weeks, was applied in mice [155]. This was applied for 14 consecutive days along with a chronic immobilization stress protocol (2 h/day) to develop anxiety-like behavior, as determined using the EPM. The authors found that exercise had an anxiolytic effect on the trained mice. Control mice showed a deficiency in 5-HT_1A_ and in the cAMP/PKA/CREB cascade in the hippocampus. Deficits in these cascades were overcome by exercise, suggesting that chronic exercise may ameliorate impaired hippocampal 5-HT_1A_-regulated cAMP/PKA/CREB signaling in an anxious brain.

Interestingly, 5HT_1A_R has also been investigated as a mediator of the anxiolytic effects of exercise in rat pups exposed to prenatal stress. Pregnant rats were exposed to a predator in a closed room to induce stress, and anxiety in the pups was measured using the EPM test [156]. Postnatal exercise in pups and non-prenatal exercise in mother rats had an anxiolytic effect on the pups, as well as increased 5HT_1A_R expression in the DRN. Consistent with these findings, in a rat model of schizophrenia, the 5-HT_1A_R antagonist WAY-100635 was observed to block the enhancement of cognitive deficits in novel object recognition elicited by acute treatment with lurasidone [157]. In conclusion, studies in other rodent models, with specific tests and adding exercise to the procedure, are necessary.

The long-term effects of aerobic exercise on BDNF-5-HT systems and cognitive function were investigated in rats of different ages. Middle-aged (8 months) and elderly (18 months) sedentary, aerobically exercised rats were studied using a lifetime moderate-intensity aerobic training program. The studies involved biochemical, immunohistochemical, and behavioral assays. The levels and expression of BDNF, 5-HT, serotonin transporter (SERT), and 5-HT receptor 1A were determined in brain areas involved in memory and learning. Cognitive function was assessed using the object recognition test. The results indicated that exercise improved age-modulated spatial and non-spatial memory systems. This result was temporally correlated with a significant upregulation of cortical, hippocampal, and striatal BDNF levels in parallel with an increase in the number of mature hippocampal CA1 neurons. Aerobic exercise also increased 5-HT levels in the brain and raphe, as well as SERT and 5-HT_1A_ receptor expression in the cortex and hippocampus. Old aerobic exercise rats showed a highly conserved response, indicating a remarkable protective effect of exercise on both systems. In summary, lifelong aerobic exercise positively affects BDNF-5-HT systems. It improves cognitive function and protects the brain against the harmful effects of a sedentary life and aging [29].

The evidence reviewed thus far utilized different types of exercise modalities. Recently, it has been investigated which exercise modality exerts the greatest anxiolytic effects in rats [158]. The exercise modalities studied were voluntary exercise (free running on a wheel), voluntary limited exercise (wheel available for only 1 h per day) or forced exercise (running on a motorized wheel for 4 weeks). The anxiolytic effects were measured through the open field test, as well as determining brain levels of 5-HT and its metabolite 5-hydroxyindoleacetic acid (5-HIAA) in the major serotonergic neuronal cell bodies and projection areas. The authors observed that the 5-HT and 5-HIAA levels in the dorsal and median raphe nuclei (MRN) increased only in the voluntary exercised group. In addition, in the paraventricular hypothalamic nucleus and in the caudate putamen, 5-HT levels increased only in the voluntary exercised group. Finally, anxiety-like behavior was reduced only in the group that exercised voluntarily. Interestingly, in the amygdala, only 5-HIAA levels were significantly increased in the voluntary exercise group. In contrast, rats in the forced exercise group were found to show no significant changes in 5-HT and increased anxiety-like behavior. The voluntary limited exercise group had no significant beneficial effects on any of the experimental parameters.

Thus, it remains to be tested whether the post-exercise modality impacts 5-HT and 5-HIAA levels specifically in areas associated with anxiety and episodic memory, a requisite for further exploring the modulation of the serotonergic system in the effects of exercise on anxiety and episodic memory. 

Interestingly, the study of kynurenine as a mediator of the protective effects of exercise is growing. In a study conducted by Laugeray et al. [159], evidence was found pointing to a pivotal role of the peripheral kynurenine pathway in modulating anxious and depressive behaviors in mice, with a focus on individual differences. These findings highlight the importance of understanding the involvement of the kynurenine pathway in mood disorders. Additionally, a systematic review by Lim et al. [160] suggests that exercise may influence kynurenine/tryptophan metabolism and psychological outcomes in individuals with age-related diseases. This indicates the potential of exercise to impact kynurenine metabolism and, therefore, affect psychological well-being.

In another study conducted by Agudelo et al. [161], it was demonstrated that the protein PGC-1α1 in skeletal muscle regulates kynurenine metabolism and plays a significant role in resilience to stress-induced depression. This study revealed that PGC-1α1 induces the expression of the enzyme kynurenine aminotransferase (KAT) in skeletal muscle, thereby controlling the balance of kynurenine/quinolinic acid in plasma and the brain. Additionally, it was observed that physical exercise activates the PGC-1α1:PPARα/δ:KAT pathway in skeletal muscle in both humans and mice.

By integrating the findings from these studies, it becomes evident that the kynurenine pathway, its metabolism through the 5-HT system, and exercise are interconnected, and may offer protection against anxiety. The work of Laugeray et al. emphasizes the role of the peripheral kynurenine pathway in mood disorders. Meanwhile, the systematic review by Lim et al. highlights the impact of exercise on kynurenine/tryptophan metabolism and psychological outcomes. Furthermore, Agudelo et al. shed light on the role of PGC-1α1 in skeletal muscle in regulating kynurenine metabolism and its association with resilience to stress-induced depression. By activating the PGC-1α1:PPARα/δ:KAT pathway, exercise may contribute to maintaining a balanced kynurenine metabolism, thus potentially offering protection against anxiety. However, despite the revised evidence, this field is still fertile for exploration as we still do not know in depth the mechanisms underlying this phenomenon.

Taken together, the evidence indicates candidate serotonergic modulators of the effects of exercise on episodic memory and anxiety, which also appear to have a specificity of expression, structure, and function (Figure 1).

**Table 2 ijms-24-11372-t002:** Preclinical studies that have investigated the effects of exercise on anxiety and memory through the serotonergic system. N.D., not determined.

Type of Exercise	Effect on Memory by the 5-HT System	References	Effect on Anxiety by the 5-HT System	References
Aerobic	Improvement	[29]	Anxiolytic	[137,138,143,144,147,151,153,155,156,158]
Resistance	N.D.		N.D.	

## 8. Conclusions

In most studies that have examined the effects of acute moderate-intensity aerobic exercise or moderate-intensity aerobic training, improvements in episodic memory and reductions in anxiety have been observed in both humans and animal models. Furthermore, the evidence suggests that this type of exercise enhances episodic memory performance, particularly in relation to its event-related and context-related components. The event component is typically evaluated through wordlist memory, while the “where” component involves the ability to recognize patterns and novel objects and places. However, the temporal component (or “when” component) has not been extensively explored, and further investigations should incorporate tests that assess the temporal aspect of episodic memory in order to draw more conclusive conclusions. This applies to both human and animal studies, provided that the specific research objectives can be accommodated.

Despite this limitation, certain findings support the notion that the effects of exercise on episodic memory are associated with functional connectivity between the DG/CA3 areas of the hippocampus and cortical regions such as the parahippocampal, angular, and fusiform gyri. Additionally, mediators of hippocampal neurogenesis, including BMP signaling, BDNF expression, and dorsal hippocampal CREB appear to play important roles in the effects of exercise on episodic memory. Similarly, the activation of microglia through the expression of BDNF and mitogen-activated protein kinase phosphatase-1 (MKP-1) in the hippocampus seems to contribute to these effects, although further research is needed to fully understand the role of MKP-1. Furthermore, these effects seem to extend to other types of memory, such as spatial memory. However, other modulatory pathways involved in these effects have been less explored. Notably, the release of the exercise-induced myokine FNDC5/irisin and the signaling of the TNF receptor through TNFR1 and TNFR2 have shown some specificity in mediating the effects of exercise on recognition memory, as well as the expression of SIRT1.

Increasing evidence demonstrates the relationship between episodic memory and anxiety. In addition, moderate-intensity aerobic exercise has been shown to improve episodic memory with anxiolytic effects. Recent research has shown that the serotonergic system seems to be a modulator of these effects. The structure specificity and the expression of specific serotonergic receptors in modulating the effects of exercise on anxiety and episodic memory appear to be of particular importance. Specifically, exercise promotes an increase in 5-HT and TPH expression in the DRN, whereas a decrease in 5-HT_2C_R mRNA expression is found in BLA and DS. These changes are related to the anxiolytic effects of exercise, indicating that these serotonergic structures and receptors are sensitive to exercise and play a role in modulating its effects on anxiety. The blockade of 5-HT_2C_R has been shown to improve recognition memory and even counteract performance deficits in animal models. It is interesting to project future research assessing whether these structures and receptors are also specifically implicated in the effects of exercise on both anxiety and episodic memory.

Similarly, exercise seems to generate anxiogenic effects by increasing 5-HT_2A_R synaptic density in BLA. There is evidence that recognition memory is enhanced when stimulated by selective agonists. In turn, it is interesting to note that the anxiolytic effects of exercise are diminished by 5-HT_2_ antagonists. Although it is not possible to infer beneficial effects on memory from the available evidence, further studies assessing its specific roles in modulating the effects of exercise in episodic memory are needed to establish conclusive relationships. In this line, it is possible that exercise may also improve anxious behavior through the increased expression of 5HT_1A_R in the DRN. In addition, its blockade has been shown to improve cognitive deficits in recognition memory tasks in models of schizophrenia. Exercise, in addition to increasing 5-HT levels in the brain and raphe, has been shown to increase SERT and 5-HT_1A_R expression in the cortex and hippocampus, improving recognition memory. Finally, only voluntary exercise, as opposed to forced exercise, has been shown to increase 5-HT and 5-HIAA levels in the DRN and MRN, as well as 5-HIAA levels in the amygdala, with anxiolytic effects.

Therefore, future research in the field should also investigate alterations in 5-HT metabolites as well as in other (largely neglected) 5-HT receptors subtypes in a detailed brain-region-specific, comprehensive manner to shed light on the exact contribution of this neurotransmitter system to the beneficial effects of physical exercise on specific cognitive domains including episodic memory and emotional regulation. Additionally, this entails further investigation into the studies regarding kynurenine, its role in mood disorders, and the influence of exercise on its metabolism. As demonstrated, the skeletal muscle protein PGC-1α1 regulates kynurenine and is associated with resilience to stress-induced depression. These recent findings suggest that exercise may provide protection against anxiety by maintaining a balanced kynurenine metabolism. However, more research is needed to fully comprehend these mechanisms. To the best of our knowledge, there are no preclinical studies that evaluate the impact of anaerobic and resistance exercise on 5-HT-mediated effects on memory and/or anxiety. Future research should also address this gap knowledge to further delineate if similar or different mechanisms underlie the mediator effect of this neurotransmitter system in the benefits of distinct exercise types on brain function.

## Figures and Tables

**Figure 1 ijms-24-11372-f001:**
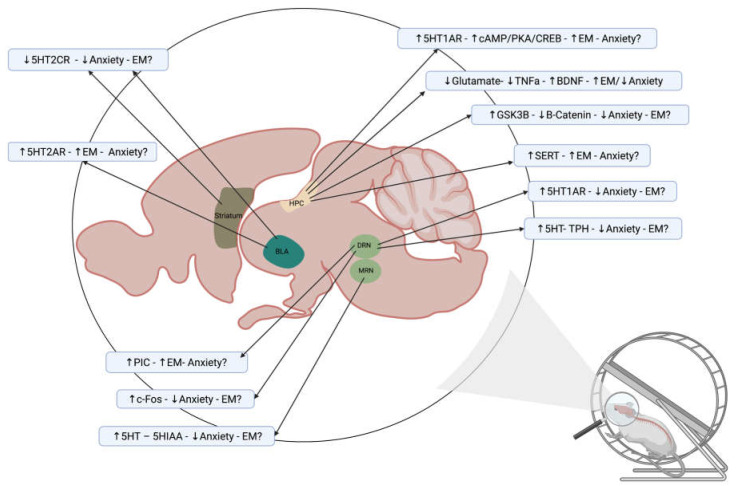
The serotonergic system as mediator of the effects of exercise on episodic memory and anxiety. Sagittal midline view of a rodent brain summarizing reported alterations in the levels of neuro-transmitter serotonin (5-HT), its metabolite (5-HIAA), synthesizing enzyme tryptophan hydroxilase (TPH), as well as in the expression of 5-HT receptors and transporter (SERT) due to exercise in animal models and the corresponding findings in episodic memory (EM) and anxiety. The arrows indicate increases or decreases; the symbol “?” indicates a lack of evidence. Key molecules/pathways re-ported in these studies are also included. Dorsal Raphe Nucleus (DRN), Medial Raphe Nucleus (MRN), Hippocampus (HPC), Amygdala (BLA), and Persistent Inward Currents (PICs) [29,117,137,138,145,147,151,155,156,158]. Created with BioRender.com.

## Data Availability

Not applicable.

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
