# Peer review of "Implications of Physical Exercise on Episodic Memory and Anxiety: The Role of the Serotonergic System"

_ijms, 2023, doi:10.3390/ijms241411372_

Round 1

Reviewer 1 Report

I suggest to restructure the manuscript and follow a better story line. At this level the manuscript present a sort of findings that most of the times the authors don't correlate one to each other sequentially and makes it difficult to follow.

I suggest to include tables, on the types of exercises and effects, break down sections on human studies and animals, and make a coherent order or the several points addressed in the manuscript.

Include a section with the Kynurenin pathways, as it is important for serotonin, and for exercise and Muscle molecular communication pathway to the brain related to stress and exercise induced-resilience to stress. i.e (Agudelo et al 2014) 

There are some typos (?) in the Figure. Add also all the abbreviations to the figure legend. 

The conclusions should mention more on point 2, as it is mostly mentioning point 1 and point 3.

The authors should also state why they focus in episodic memory, why it is more important than others or not, why this and not other types of memory?

The authors made some comments on inflammatory parameters, although this is not the scope of the review, but actually the ones that are mentioned are not the most relevant as it should be also mentioned IL6 which is a myokine and highly relevant during exercise.

The MS should be revisited by an expert in English language

Author Response

We thank Reviewer 1 for the thorough revision and suggestions provided, which undoubtedly have improved our manuscript. Please find replies point-by-point below.

I suggest to restructure the manuscript and follow a better story line. At this level the manuscript present a sort of findings that most of the times the authors don't correlate one to each other sequentially and makes it difficult to follow.

R: As suggested by the rewiever, we  have restructured the whole manuscript and improved the story line. 

I suggest to include tables, on the types of exercises and effects, break down sections on human studies and animals, and make a coherent order or the several points addressed in the manuscript.

R: As suggested by the reviewer, the manuscript now has Tables to facilitate the structure and order of the reviewed literature.

Include a section with the Kynurenin pathways, as it is important for serotonin, and for exercise and Muscle molecular communication pathway to the brain related to stress and exercise induced-resilience to stress. i.e (Agudelo et al 2014) 

R: As suggested by the reviewer, Section 7 (lines 2333-2370) now includes a discussion about kynurenin pathway.

There are some typos (?) in the Figure. Add also all the abbreviations to the figure legend. 

R: We have corrected the Figure. The symbol "?" represents lack of evidence on the corresponding behavioral output, i.e. anxiety or episodic memory.  Missing abbreviations are now listed.

The conclusions should mention more on point 2, as it is mostly mentioning point 1 and point 3.

R. We have improved the conclusions section to address this.

The authors should also state why they focus in episodic memory, why it is more important than others or not, why this and not other types of memory?

R: Disorders affecting episodic memory can lead to amnesic syndromes in humans, which is particularly evident in  Alzheimer's Disease and conditions such as hippocampal sclerosis, as well as in depression and PTSD. The interplay between anxiety and episodic memory, as well as the impact of exercise on episodic memory has been much less investigated compared to other memory systems. In the newly structured manuscript, sections 2 to 5 have been improved to clarify this. 

The authors made some comments on inflammatory parameters, although this is not the scope of the review, but actually the ones that are mentioned are not the most relevant as it should be also mentioned IL6 which is a myokine and highly relevant during exercise.

R: Inflammatory patterns are indeed beyond the scope of this review.

Reviewer 2 Report

In my opinion, the review manuscript entitled “Implications of physical exercise on episodic memory and anxiety: the role of the serotonergic system” is very informative as well as generally well organized and well-written. Undoubtedly, the chosen subject of the paper is highly relevant. I have only two minor comment:

* Tables summarizing in vivo (pre-clinical and clinical) studies supporting beneficial effects of exercise on episodic memory and anxiety should be added.

* References published before 2000 should be updated.

English language is fine.

Author Response

We thank reviewer 2 for the thorough revision and comments provided. Please find  point-by-point responses below.

In my opinion, the review manuscript entitled “Implications of physical exercise on episodic memory and anxiety: the role of the serotonergic system” is very informative as well as generally well organized and well-written. Undoubtedly, the chosen subject of the paper is highly relevant. I have only two minor comment:

  • Tables summarizing in vivo (pre-clinical and clinical) studies supporting beneficial effects of exercise on episodic memory and anxiety should be added.

R: As requested by the reviewer, we have included tables that allow to better structure and organize the revised literature. 

  • References published before 2000 should be updated.

R: References have been updated. 

Round 2

Reviewer 1 Report

The manuscript has improved very much.

However, the tables are still very vague and must improve. I suggest to depict and specify the type of effect seen for anxiety (i.e anxiogenic or anxiolytic) and the type of effect seen in the episodic memory (Increase/decrease) and then the references next. The same for table 2. 

These improvements will make it more clear for the reader and improve the quality of the review manuscript.

Author Response

We thank Reviewer 1 for the comments and suggestions provided. As requested, we have improved Tables 1 and 2 specifying the reported effects of exercise on anxiety and episodic memory with the corresponding references.